# Real World Practice Study of the Effect of a Specific Oral Nutritional Supplement for Diabetes Mellitus on the Morphofunctional Assessment and Protein Energy Requirements

**DOI:** 10.3390/nu14224802

**Published:** 2022-11-13

**Authors:** Juan J. López-Gómez, Cristina Gutiérrez-Lora, Olatz Izaola-Jauregui, David Primo-Martín, Emilia Gómez-Hoyos, Rebeca Jiménez-Sahagún, Daniel A. De Luis-Román

**Affiliations:** 1Servicio de Endocrinología y Nutrición, Hospital Clínico Universitario de Valladolid, 47003 Valladolid, Spain; 2Centro de Investigación Endocrinología y Nutrición, Universidad de Valladolid, 47002 Valladolid, Spain

**Keywords:** diabetes, prediabetes, oral nutritional supplement, enteral nutrition, morphofunctional assessment

## Abstract

Introduction: The prevalence of malnutrition in patients with diabetes mellitus is high. In these patients, monitoring nutritional intervention is complex. Aims: To evaluate the evolution in the nutritional status in patients with diabetes/prediabetes and malnutrition with a diabetes-specific enteral formula. Methods: Real-life study of one arm in 60 patients with diabetes and prediabetes, performing a dietary adaptation with diabetes-specific oral nutritional supplementation. A morphofunctional assessment was performed, consisting of intake assessment, anthropometry, body composition (bioimpedance and muscle ultrasound), handgrip strength and biochemical markers. The diagnosis of malnutrition was made using the criteria of the Global Leadership Initiative on Malnutrition (GLIM). The variables were measured at baseline and 3 months after starting the intervention. Results: The mean age was 67.13 (14.9) years. In total, 30 (50%) of the patients were women. Of the total, 60% of the patients had diabetes mellitus and 40% of the patients had prediabetes. The initial body mass index was 24.65 (5.35) kg/m^2^. It was observed that 80% of the patients had malnutrition, whereas after the intervention, the prevalence was 51.7% (*p* < 0.01). At the beginning of the study, 20% of the patients suffered from sarcopenia and after the intervention it was 16.7% (*p* = 0.19). Conclusions: Medical Nutrition Therapy with an adapted oral diet associated with diabetes-specific oral nutritional supplementation reduces malnutrition in patients at nutritional risk and disturbances of carbohydrate metabolism.

## 1. Introduction

Disease-related malnutrition (DRM) is a pathology with a high prevalence, reaching up to 60% in hospitalized patients with chronic diseases [1]. This malnutrition is more striking in elderly patients and is closely related to sarcopenia, another highly prevalent disease in elderly patients.

Recently, it is being postulated that diabetes mellitus may be a factor favoring malnutrition and sarcopenia. In fact, in institutionalized diabetic patients over 65 years, it has been observed that 21.2% are malnourished and that 39.1% are at risk of malnutrition [2]. This can be related to two situations that occur in diabetic patients: First, there is a sustained metabolic alteration that makes it difficult to manage energy properly, especially carbohydrates. This circumstance promotes a prooxidative state that increases the risk of chronic complications and produces a deterioration in muscle mass and a worser nutritional status. Furthermore, the use of nutrient-restrictive diets is a risk factor for causing imbalances in energy balance. In fact, in the study carried out by Serrano-Valles et al., it was observed that patients with type 2 diabetes mellitus had a worser nutritional situation than patients without diabetes, and this situation was associated with a longer hospital stay [3]. On the other hand, an increase in sarcopenia [4] and sarcopenic obesity [5] has been observed in patients with diabetes mellitus, which conditions a decline in muscle strength and functionality and is associated with a worsening of quality of life of the patient and an increase in mortality [6]. Moreover, glycemic control correlates with muscle mass and function [7].

The diagnosis of malnutrition is difficult because it does not depend only on the weight at a given time, but also on its evolution and the underlying pathological situations [8]. Classically, body mass index has been used as a measure of the patient’s nutritional status, but this measure is not the most appropriate and has evident limitations in different pathologies that can make it possible to maintain an adequate weight with a deterioration of the “metabolically active mass” [9]. These pathologies can produce an increase in fat mass (obesity) or body water (heart failure, liver failure, kidney failure) [10]. Therefore, the clinical use of body composition measurements is essential for adequate assessment of this malnutrition, especially in the evaluation of muscle mass and function.

In this context, nutritional assessment can no longer be based on the determination of anthropometric measurements. The concept of morphofunctional assessment postulates that the diagnosis and monitoring of nutritional status must be carried out using techniques that determine the evaluation of intake, anthropometry, body composition, muscle strength and function. This new concept of nutritional evaluation should be implemented in the clinical management of the patient and in the determination of variables in clinical research in nutrition [11,12].

The prevalence of malnutrition in patients with diabetes mellitus and the difficulty in assessing it is high due to the limitations of classical techniques such as body mass index. On the other hand, monitoring nutritional improvement in these patients is complex in relation to the above-mentioned reasons. In this context, we need to deploy a comprehensive assessment of nutritional status. This type of assessment combines methods of determining body composition and muscle strength according to the concept of the Morphofunctional Assessment of Disease-Related Malnutrition. This new concept can allow us to obtain more valuable information in the diagnosis and evolution of the patient in Medical Nutrition Therapy [10].

Intervention studies with specific nutritional oral supplementation in diabetes are scarce and conducting clinical trials of intervention is difficult due to ethical problems in the comparative arm. This means that real life studies can provide us with additional information. These studies could generate evidence obtained from routine clinical practice data. This is the principal value from data obtained outside the context of randomized controlled trials [13].

In patients with diabetes and malnutrition, nutritional intervention for caloric increase can be associated with a worsening of glycemic control. This is the reason to use diabetes-specific oral nutritional supplements. These formulas can have the following characteristics: (a) the reduction of energy from carbohydrates and replace it with energy from lipids; (b) use of carbohydrates with a low glycemic index such as lactose or isomaltulose; or (c) increase the amount of soluble fiber to decrease glucose absorption [14]. These types of formulas usually have a high percentage of protein to improve nutritional status and muscle function. In addition, these formulas are usually enriched in monounsaturated (MUFA) and polyunsatured fatty acids (PUFA) and lead to finding benefits in the lipid profile of these patients [15].

For this reason, a real-life study is proposed to describe the effect of diabetes-specific oral nutrition supplementation in patients with disease-related malnutrition. The main objective of this study was to prove the influence of Medical Nutrition Therapy with a specific oral nutritional supplementation through morphofunctional assessment in patients with malnutrition and diabetes or altered metabolism of carbohydrates.

## 2. Materials and Methods

### 2.1. Design

This is an open-label, prospective, interventional study. In this study, the nutritional status of diabetic patients and their evolution was evaluated. The evolution was based on nutritional measurements performed according to medical nutrition therapy (adapted dietary recommendations and diabetes-specific oral nutritional supplementation). It was proposed as a real-world study with data obtained from routine clinical practice.

An exhaustive anamnesis was carried out on affiliation data, personal history, evolution of the disease and nutritional history. Classic anthropometric evaluation, bioelectrical impedanciometry and muscle ultrasound evaluation was performed. Nutritional parameters were measured according to usual clinical practice.

Oral nutritional supplementation was started with a specific normocaloric and hyperproteic formula for diabetic patients in routine clinical practice. The medical and nutritional treatment prescribed at the initial visit and during follow-up was recorded. Records of the evolution of the morphofunctional assessment (anthropometry, nutritional ultrasound, electrical and biochemical bioimpedance analysis) were taken at the beginning and 3 months after the start of nutritional support.

The study was carried out in accordance with the Declaration of Helsinki and all procedures were approved by the Medical Research Ethics Committee (CEIm) of the Hospital Clínico Universitario de Valladolid under code PI 20-1967.

### 2.2. Study Subjects

The study was developed in patients with malnutrition referred to the Clinical Nutrition consultation of the East Valladolid Area. Patient recruitment was carried out between January 2021 and September 2022.

The patient inclusion criteria were patients with diabetes mellitus or prediabetes at risk of malnutrition, and need for specific oral supplementation of diabetes mellitus and age over 18 years. The exclusion criteria were decompensated liver disease; chronic kidney disease stage IV or higher; inability to walk; and non-signing of the informed consent by the patient.

### 2.3. Nutritional Intervention

The patients received the following nutritional education and medical nutrition therapy:Patients received education on adapted oral diet to increase calories and protein in patients with diabetes or carbohydrate metabolism disorders (prediabetes).Patients received nutritional education with a dietitian in adaptation of oral diet to increase protein–energy intake and they received education in consumption of oral nutritional supplementation. The adherence of these diets was assessed every fourteen days with a phone call by a dietitian to improve the calorie restriction and macronutrient distribution. The diet compliance was verified with a telephone nutritional questionnaire every fourteen days and a four-day nutritional questionnaire during face-to-face visits.Oral nutritional supplementation with a hyperproteic normocaloric formula specific for diabetes (carbohydrates with a low glycemic index, insoluble fiber) (Nutavant Plus Diabetica^®^) (Table 1). The amount (1 or 2 bottles) was adjusted according to the nutritional requirements of the patient and the estimation of usual intake [16,17].

### 2.4. Study Variables

Clinical variables: Age (years); gender (male/female); systolic and diastolic blood pressure (mmHg); presence of concomitant pathologies.Anthropometry: The anthropometric variables measured were weight (kg); height (meters); body mass index (BMI) (weight/height × height) (kg/m^2^); arm circumference (AC); and calf circumference (CC). The percentage of weight loss was calculated: Start Weight Loss = ((Usual weight (kg) – Present weight (kg))/Usual weight) × 100; and 3 Months Weight Loss = ((Initial weight (kg) − 3 months weight)/Initial weight) × 100.Biochemical variables: They were performed with a Cobas c-711 autoanalyzer (Roche Diagnostics): Glucose (mg/dL); total cholesterol (mg/dL); HDL cholesterol (mg/dL); LDL cholesterol (mg/dL); triglycerides (mg/dL); albumin (g/dL); HbA1c (%), C-Reactive Protein (CRP) (mg/dL), prealbumin (mg/dL); and CRP/prealbumin ratio.Energy Expenditure and Nutritional Requirements: The energy expenditure of the patients was determined by means of the Harris–Benedict Equation multiplied by a Stress Factor of 1.3 and the protein requirements were determined by means of the factor 1–1.5 g of protein per kilogram of the patient’s adjusted weight. We based the requirements on the patient’s clinical situation and comorbidities as the recommendations made by the clinical guidelines of the European Society for Clinical Nutrition and Metabolism in surgery and oncology suggests. This decision was made because most of the patients had underlying oncological and/or surgical pathology [18,19].Nutritional questionnaire: All subjects completed a 4-day prospective nutritional questionnaire to assess calorie and macronutrient intake. This questionnaire was conducted before starting the intervention and 3 months after its start. The importance of not modifying dietary habits was insisted on so that it would be representative. All study participants were instructed to record food intake, daily and prospectively, with the help of food scales to facilitate precision in portion sizes. They were also asked about the way of preparing said foods. Records were reviewed by a dietitian and analyzed by a Dietsource^®^ data processing computer system (Nestle, Geneve, Switzerland). Total calorie intake was used as an indicator of nutritional intake. No subject was taking dietary supplements or following any type of diet at the start of the study or in the 6 months prior to the study. Nutritional intake was measured in absolute values (in kilocalories (kcal) or grams (g)) and in percentages of the total caloric value. The nutritional questionnaire assessed the total energy intake, measured in kilocalories, as well as the different macronutrients: proteins, carbohydrates, fats and fiber, all of them measured in grams. The amount of protein ingested per kilogram of body weight was also calculated.Muscle functionality variables: Hand dynamometry (JAMAR^®^ dynamometer): non-dominant hand dynamometry was performed with the patient seated and the arm at a right angle to the forearm. Three measurements were made and the average of the three measurements was made. The diagnostic criteria of low muscle strength proposed by the European Working Group on sarcopenia in older people (EWGSOP2) [20] were used. (<27 kg in men and <16 kg in women).Corporal Composition:

Bioimpedanciometry (BIA 101 Anniversary; EFG Akern): The BIA was performed between 8:00 and 9:15 h, after an overnight fast and after a time of 15 min in the supine position. The BIA measured the geometrical components of impedance (Z), resistance (R) and the capacitance component (X). The PhA is derived for the following equation PhA = (X/R) × (180°/π). The BIA provided data regarding fat mass (FM), fat-free mass (FFM), skeletal muscle mass (SMM), fat free mass index (FFMI) and percentage of skeletal muscle mass (%MM) [14]. All these data are based on raw electrical data from BIA multifrequency at 50 Hz [14].

Skeletal Appendicular Mass Index (ASMI): ASMI (kg/m^2^) was estimated by bioimpedanciometry applying Sergi’s formula [19]: −3.964 + (0.227 × RI) + (0.095 × weight) + (1.384 × sex) + (0.064 × Xc), where RI is Resistivity Index and Xc is reactance (sex: Male = 1; Female = 0).

European Working Group on Sarcopenia in Older People (EWGSOP2) diagnostic criteria of sarcopenia for low muscle mass (ASMI < 7 kg/m^2^ in men and ASMI < 5.5 kg/m^2^ in women) were used [21].

Muscle ultrasound (Mindray Z60): Muscle ultrasound of the quadriceps rectus femoris (QRF) of the left and right lower extremities with a 10 to 12 MHz probe and a multifrequency linear matrix (Mindray Z60, Madrid, Spain) were performed in all subjects (patient in supine position). The probe was aligned perpendicularly to the longitudinal and transverse axis of the non-dominant QRF. The evaluation was performed without compression at the level of the lower third from the superior pole of the patella and the anterior superior iliac spine, measuring the anteroposterior muscle thickness, circumference and cross-sectional area [17]. The measurements made using this technique were: muscle area (cm^2^) (MARA) and the index of the muscle area with respect to height (cm^2^/m^2^) (MARAI), the X-axis of QRC (cm), Y-axis of QRC (cm) and X/Y index [12].

Malnutrition and Sarcopenia diagnosis: The diagnosis of malnutrition was made using the Global Leadership Initiative on Malnutrition (GLIM) criteria, using the ASMI estimated by bioimpedance measurement measured by impedance measurement as an evaluation variable for muscle deterioration (ASMI muscle mass reduction < 7 kg/m^2^ in men was considered and <5.5 kg/m^2^ in women) [8]. On the other hand, the diagnosis of sarcopenia was made according to the revised criteria for sarcopenia of the EWGSOP2, using the ASMI estimated by bioimpedance as a determination of decreased muscle mass with handgrip strength to estimate the function to diagnose sarcopenia [20].

### 2.5. Data Analysis

The data was stored in a database of the statistical package SPSS 23.0 (SPSS Inc., Chicago Illinois, USA) with an official license from the University of Valladolid. A normality analysis of continuous variables was performed with the Kolmogorov-Smirnov test.

Continuous variables were expressed as mean (standard deviation). The difference in means between parametric variables was analyzed with the unpaired and paired t-Student test, and the non-parametric variables with the Mann–Whitney U-test and the Kruskal–Wallis K-test. An intention-to-treat analysis of patients who consumed supplementation more than a half time of intervention was conducted. A significant difference was considered as a *p*-value of less than 0.05.

## 3. Results

A total of 75 patients were recruited, of whom 60 (80%) were analyzed (Figure 1). In total, 36 (60%) patients had a diagnosis of diabetes mellitus and 24 (40%) patients suffered from some disorder of carbohydrate metabolism without a diagnosis of diabetes mellitus (altered fasting blood glucose, intolerance to carbohydrates, glycated hemoglobin in prediabetes range (5.7–6.4%).

Of the total number of patients, 30 (50%) patients were women, and 30 (50%) patients were men. The mean age of the patients was 67.13 (14.09) years.

Most of the patients suffered from oncological pathology (68.3%%); the rest of the patients suffered from cardiopulmonary pathology (10%), non-oncological digestive pathology (13.3%), and neurological (1.7%) and other pathologies (6.7%).

### 3.1. Sample Description

After performing the nutritional assessment, data were obtained from anthropometry, dynamometry, electrical bioimpedance measurement, and muscle ultrasound (Table 2). According to the GLIM criteria, 48 (80%) patients suffered from malnutrition and according to the EWGSOP2 criteria, 12 (20%) patients presented sarcopenia.

After carrying out the initial analysis of the diet, a consumption below the caloric-protein requirements was observed (Table 3).

The metabolic parameters of the patients prior to the intervention were evaluated. An increase in glycated hemoglobin, glucose and triglycerides was observed in patients with diagnosed diabetes mellitus compared to those patients with carbohydrate alterations (Table 4).

### 3.2. Nutrional Therapy Intervention

Nutritional supplementation was started with a specific hyperproteic normocaloric diabetes formula. It was prescribed based on the requirements and the calculated dietary intake. A bottle (250 mL) was consumed by 44 patients (73.3%), two bottles by 15 patients (25%) and half a bottle (125 mL) by 1 patient (1.7%). At three months from the beginning of intervention, 56 (93.3%) patients consumed 100% of oral nutritional supplementation prescribed, 1 (1.7%) consumed 50%, 1 (1.7%) consumed 25% and 2 (3.3%) patients consumed no supplementation.

Influence of the intervention on intake

A significant increase in caloric intake (Baseline: 1364 (417) kcal/day; 3 months: 1666 (519) kcal/day; *p* < 0.01) and of all nutrients (proteins, fats and carbohydrates) was observed, with an increase in the percentage of carbohydrates and a decrease in fat over the total caloric value (Figure 2).

An improvement in caloric adjustment to the percent of protein and energy requirements was observed. On the other hand, an increase in protein consumption per kg of weight was observed (Initial: 1.14 (0.43) g/kg/day; 3 months: 1.38 (0.49) g/kg/day; *p*-value < 0.01) (Figure 3).

An improvement in consumption of monounsaturated fatty acids (MUFA) and polyunsaturated fatty acids (PUFA) after the intervention was observed (Table 4). In the same way, an improvement in consumption of minerals except sodium and copper was noted, as was a consumption of vitamins except vitamin A, B1, B3, B12, C and D (Table 5).

Influence of the intervention on body composition

A decrease in the percentage of weight loss was observed at 3 months in the total sample and stratified according to sex. No deterioration in anthropometric parameters (weight, muscle circumference, calf circumference), body composition (bioimpedanciometry and muscle ultrasound) or muscle function (hand dynamometry) was observed after the start of oral nutritional supplementation. No difference was observed according to sex (Table 6).

A significant decrease in the malnutrition (GLIM criteria) rate was observed 3 months after the intervention. However, a slight, non-significant decrease in the prevalence of sarcopenia (EWGSOP2) was observed in the sample studied (Figure 4).

In patients with sarcopenia, an improvement in muscle strength measured by dynamometry in the non-dominant hand was observed 3 months after the start of the nutritional intervention (Baseline: 9.83 (5.49); 3 months: 11.33 (6.11), *p*-value = 0.04).

Influence of the intervention on biochemical parameters

There were no baseline differences in biochemical parameters based on gender. The evolution in the biochemical parameters was analysed according to the presence or not of diabetes mellitus due to the baseline differences of the groups at this level.

A significant increase in glycated haemoglobin was observed in patients with diabetes mellitus, although it was not observed in patients with alterations in carbohydrate metabolism. On the other hand, a significant increase in plasma albumin was observed in patients with diabetes mellitus (Table 7).

## 4. Discussion

The use of medical nutrition therapy with an adapted oral diet and diabetes-specific supplementation in patients with high nutritional risk and alterations in carborhydrates metabolism (diabetes mellitus and prediabetes) was related to an achievement of nutritional requirements (calorie-protein) in our study. This was associated with cessation of previous weight loss and maintenance of morphofunctional assessment parameters (anthropometry, body composition and muscle strength).

The patients analyzed were referred to the Clinical Nutrition consultation as they were in a situation of nutritional risk. Most of the patients in this sample suffered from cancer. This type of pathology is associated with an increased risk of malnutrition. It has been observed that between 15 and 40% of cancer patients present some degree of malnutrition at diagnosis of the disease [22]. On the other hand, these pathologies are associated with alterations in carbohydrate metabolism in relation to the disease itself or its treatment (chemotherapy, corticosteroids, etc.). This circumstance can be associated with a decreased diagnosis of the state of malnutrition due to the observation of high body mass indexes; and, in addition, it can be associated with a tendency to carry out a dietary restriction to control the metabolic complications of diabetes mellitus [3]. All this can enhance the state of malnutrition by not carrying out adequate medical nutrition therapy.

Most of the patients had malnutrition according to GLIM criteria, and this circumstance can be considered normal given that the patients had been referred for nutritional assessment as they were at high nutritional risk. However, the rate of sarcopenia was quite high (20%) since the population was not an elderly population. Other series of patients with diabetes mellitus have also shown high prevalence of muscle mass deterioration, such as the study by Park et al. which showed a more striking deterioration of skeletal muscle mass in patients with diabetes mellitus [23]. In fact, it has been observed that there are many factors that can negatively influence muscle mass in patients with diabetes, such as poor glycemic control [7], the use of certain treatments for glycemic control that can enhance muscle loss [24] and a sedentary lifestyle, which is a risk factor for diabetes itself [25].

Differences in terms of muscle strength and body composition according to gender have been observed. This factor makes it impossible to analyze the evolution of the total sample so it is necessary to stratify results according to gender.

A decreased caloric and protein intake at baseline was observed with respect to the estimated requirements based on the clinical guidelines. The criteria used to calculate requirements was based on the ESPEN clinical guidelines for surgical and oncological patients, given the characteristics of the patients in the sample [18,19]. This decrease could be due to the underlying pathology and could be related to the high percentage of weight loss observed in our population. After the start of the nutritional intervention, the energy-protein requirements were achieved.

Supplementation was selected in relation to nutritional requirements (protein requirements above energy requirements) and specifically in diabetes mellitus due to the metabolic characteristics of the patients. Use of specific formulas for diabetes in patients with nutrition are widely studied. These types of preparations that change the amount and type of carbohydrates (low glycemic index) and lipids (predominantly monounsaturated fatty acids) have shown an improvement in glycemic control and lipid control [15,26]. However, the evidence regarding its use as a supplementation to an incomplete diet is not as well studied.

The use of this type of supplementation first showed a stabilization of weight loss in our patients. The objective of reaching the caloric-protein requirements was adequate, especially if we consider that most of these patients presented a basic oncological pathology in which there is a tendency for progressive weight loss in relation to the oncological treatment and for those who started from a baseline situation of normal weight. On the other hand, three months after the start of treatment with this type of supplementation, a decrease in the diagnosis of malnutrition was observed in the total sample; therefore, the first objective of supplementation was achieved. These objectives are like those recommended for the use of oral nutritional supplements in patients with malnutrition with oncological or surgical pathology or elderly patients according to the ESPEN recommendations [18,19].

No significant change In body com”osit’on parameters was observed, although there was a trend towards improvement in muscle mass in both men and women. On the other hand, in patients with a diagnosis of sarcopenia at the beginning of the study, it was observed that in women there was a significant improvement in dynamometry. Nutritional intervention in patients with diabetes and sarcopenia can improve this situation, as was shown in the study by Maykish et al. that evaluated the use of different branched-chain amino acids in the management of sarcopenia and their involvement in the modulation of diabetes [27]. On the other hand, patients with diabetes mellitus have a high prevalence of sarcopenic obesity that may require more adapted treatment [5].

The adequate adjustment of the diet in the patient with malnutrition is basic. If the requirements cannot be achieved with an adaptation of the diet, the use of oral nutritional supplementation is necessary to meet these requirements. In our sample, an adequate range of caloric-protein requirements was observed with the nutritional intervention. However, in patients with diabetes, it is also necessary to achieve an adequate glycemic control because it has been observed that poorer glycemic control is associated with a greater decline of muscle mass [7].

A slight increase in glycated hemoglobin and plasma albumin was observed in patients with diagnosed diabetes mellitus. This may be related to the increase in the consumption of carbohydrates after the decrease in the initial intake in relation to the underlying pathology. This data differs from other data observed in studies with diabetes-specific formulas in which an improvement in glycemic parameters was observed, although in these studies, medical nutrition therapy with complete enteral nutrition was usually evaluated [26]. In other studies, in which oral nutritional supplements were used, they were used as a substitute for meals and not in patients with malnutrition, so the results are not comparable with our population [28]. However, in the sample studied, no alterations were observed in basal glycaemia or in lipid parameters (cholesterol and tryglicerides), neither in patients with diabetes nor in patients with prediabetes. This fact could be related to the increase in lipids with MUFA and PUFA consumption despite the increase of carbohydrate consumption [29].

Albumin is an imprecise biomarker that can be interfered with in many situations, such as with inflammation and the hydration state of the patient. In this study, albumin levels were not below the lower limit of normal. The change of this parameter in our sample is unspecific and there is no easy explanation. More specific nutritional biomarkers in the PCR/prealbumin ratio did not show differences but its use is promising to evaluate the prognosis, especially in patients with acute pathologies [10].

The use of diabetes-specific oral nutritional supplementation has shown better postprandial glycemic control after its intake [14,30]. Other studies using a meal-replacement plan during a short period of time have shown an improvement in the glycemic profile [28]. These interventions were not used in patients with disease-related malnutrition. In these studies, the type of prescribed diet, the underlying disease and adherence to oral nutritional supplementation can influence the results.

The main strength of this study was the evaluation of a diabetic-specific formula as an oral nutritional supplement associated with diet in real clinical practice, since there are not many studies that evaluate this method of medical nutrition therapy. This fact allows us to extrapolate our results to generalized clinical practice. On the other hand, the evaluation from a morphofunctional point of view in thess type sof patients allows for a complete assessment and for monitoring the different spheres of nutritional status (evaluation of intake, anthropometry, body composition and muscle function) in order to personalize most appropriate way of treatment.

The limitations of this study are, first, the non-use of a control group that would allow us to evaluate the specific effect of the formula with respect to standard or other specific formulas. In addition, the selected sample has a predominance of cancer patients in whom the effect of nutrition can be variable depending on the stage and treatment of the disease. This situation may interfere with the results and would require a larger sample size to perform adequate stratification.

This study allows us to propose new lines of research on the use of diabetes-specific nutritional supplementation, with a control group and in specific groups of patients at nutritional risk. The use of the different morphofunctional assessment techniques must be basic in all nutritional assessment studies, given that we increasingly have more techniques that can be used in our daily clinical practice, such as bioimpedance measurement and nutritional ultrasonography.

## 5. Conclusions

Medical Nutrition Therapy with an adapted oral diet and oral diabetes-specific nutritional supplementation reduces malnutrition according to GLIM criteria in patients at nutritional risk with alterations in carbohydrates metabolism. The choice of a diabetes-specific formula produces a slight increase in glycated hemoglobin in patients with diabetes but without a significant alteration in the rest of the metabolic parameters. In patients with BMI in the normal range, this intervention can produce a stabilization in morphofunctional assessment parameters; and, in women with sarcopenia, it shows an improvement in muscle strength measured by hand dynamometry.

## Figures and Tables

**Figure 1 nutrients-14-04802-f001:**
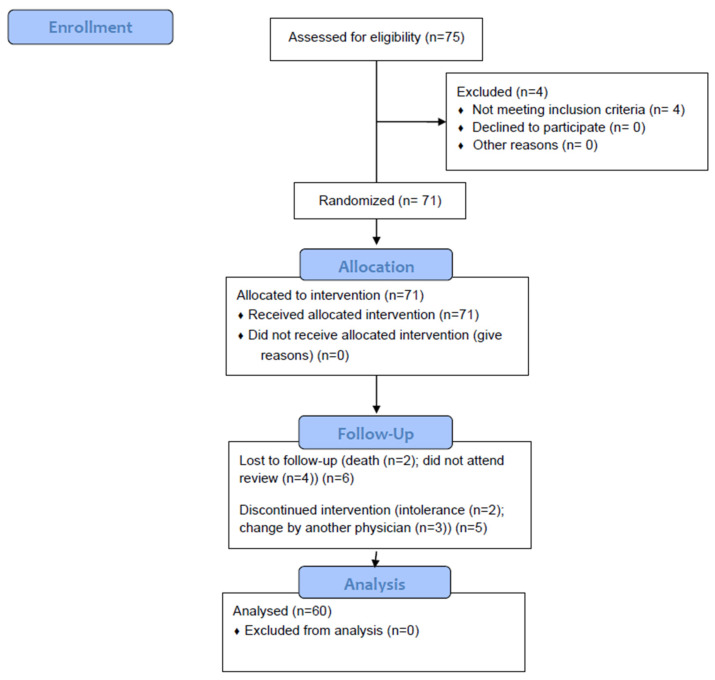
Flow chart.

**Figure 2 nutrients-14-04802-f002:**
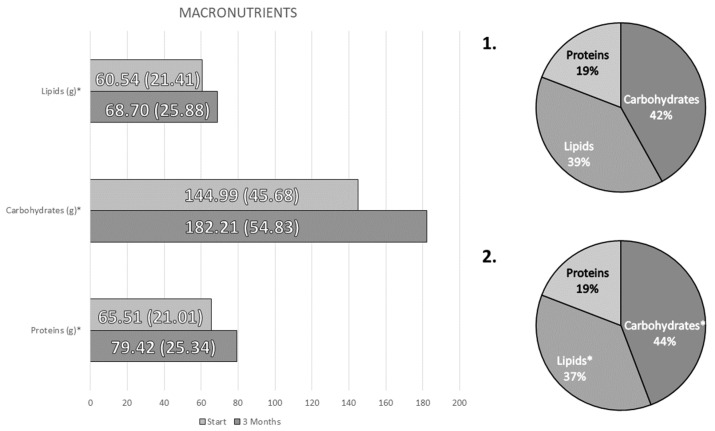
Differences in the consumption of macronutrients between the beginning of the intervention (**1**) and 3 months after it (**2**). * *p*-value < 0.05.

**Figure 3 nutrients-14-04802-f003:**
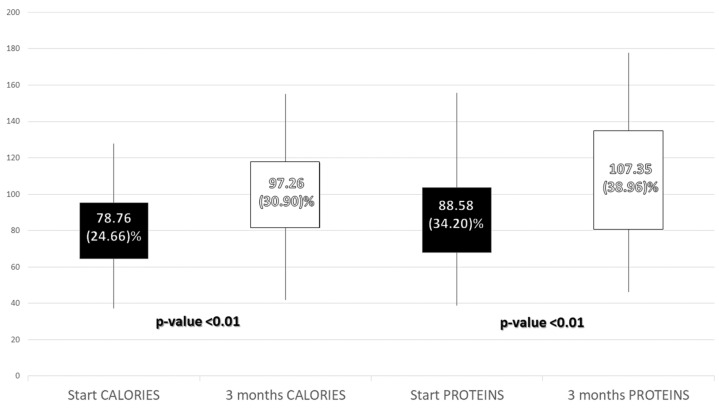
Percent of adjustment to caloric and protein requirements before and 3 months after the intervention.

**Figure 4 nutrients-14-04802-f004:**
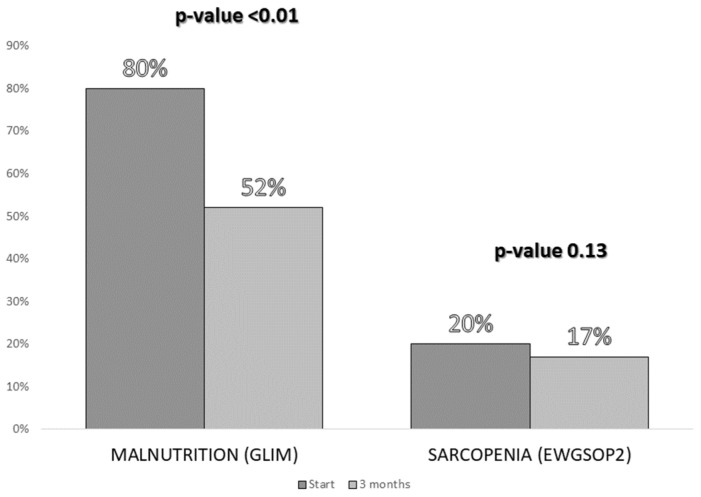
Comparison of the percentages of malnutrition (according to the Global Leadership Initiative on Malnutrition (GLIM) criteria) and sarcopenia (according to the European Working Group on Sarcopenia in Older People (EWGSOP2) criteria) before the intervention and 3 months after the start of the intervention.

**Table 1 nutrients-14-04802-t001:** Composition of Specific Diabetes Formula used as intervention.

Diabetes Specific Formula (250 mL Bottle)	
Caloric Content (kcal)	300
Proteins (g (% TCV ^1^))	17 (22.66%)
Lipids (g (% TCV))	11.7 (35.1%)
Saturated (g)	2.6
MCT (g)	1.7
MUFA (g)	5.9
PUFA (g)	2.8
w-3 (g)	0.83
w-6 (g)	1.88
Carbohydrates (g (%TCV))	30 (40%)
Sugars (g)	6.3
Isomaltulose (g)	3
Minerals	
Sodium (mg)	278
Chloride (mg)	113
Potassium (mg)	333
Calcium (mg)	275
Phosphate (mg)	238
Magnesium (mg)	50
Iron (mg)	2.8
Zinc (mg)	2
Copper (mg)	0,20
Iodine (mg)	30
Selenium (mg)	11
Manganese (mg)	0.40
Chrome (mg)	45
Molybdenum (mg)	10.6
Fluoride (mg)	0.58
Vitamins	
Vitamin A (mg)	160
Vitamin D (mg)	1.6
Vitamin K (mg)	15
Vitamin C (mg)	16
Thiamin (mg)	0.22
Riboflavin (mg)	0.28
Vitamin B6 (mg)	0.28
Niacin (mg)	3.3
Folic Acid (mg)	40
Vitamin B12 (mg)	0.50
Pantothenic acid (mg)	1.2
Biotin (mg)	10
Vitamin E (mg)	2.4
Inositol (mg)	38
Choline (mg)	38
Osmolarity (mOsm/L)	315
Fiber (g)	4.5

^1^ %TCV: Percentage Total Calorie Value.

**Table 2 nutrients-14-04802-t002:** Differences in the Morphofunctional Assessment at the beginning according to sex.

	Total	Men	Women	*p*-Value
Sarcopenia (EWGSOP2)	20%	3.3%	36.7%	<0.01
Malnutrition (GLIM)	80%	86.7%	73.3%	0.19
Diabetes Mellitus	60%	66.7%	53.3%	0.29
Age (years)	67.13 (14.9)	68.70 (12.11)	65.57 (15.89)	0.39
Anthropometry				
BMI (kg/m^2^)	24.65 (5.35)	25.53 (4.30)	23.77 (6.18)	0.20
Braquial circumference (cm)	24.71 (3.52)	25.43 (2.53)	23.99 (4.21)	0,11
Calf Circumference (cm)	31.69 (3.61)	32.75 (3.11)	30.63 (3.81)	0,02
Handgrip Strength				
Handgrip Strength (kg)	20.60 (8.26)	25.42 (7.65)	15.79 (5.67)	<0.01
Bioelectrical Impedanciometry				
Resistance (ohm)	545.4 (91.25)	502.53 (75.11)	588.27 (86.59)	<0.01
Reactance (ohm)	46.9 (9.26)	44.7 (9.11)	49.10 (9.01)	0.06
Fase Angle (°)	4.95	5.11 (0.86)	4.78 (0.66)	0.11
ASMI (kg/m^2^)	6.43 (1.11)	7.07 (0.91)	5.79 (0.91)	<0.01
FFMI (kg/m^2^)	17.46 (3.05)	18.27 (2.70)	16.65 (3.20)	0.04
FMI (kg/m^2^)	6.78 (3.32)	6.58 (2.34)	6.98 (4.10)	0.64
BCMI (kg/m^2^)	8.29 (1.66)	8.93 (1.69)	7.64 (1.37)	<0.01
%TBW	56.17 (8.90)	58.60 (4.49)	53.75 (11.35)	0.03
Rectus Femoris Ultrasonography				
RFAI (cm^2^/m^2^)	1.27 (0.47)	1.36 (0.55)	1.17 (0.35)	0.14
X/Y (cm^2^/m^2^)	3.59 (1.57)	3.39 (1.56)	3.79 (1.57)	0.34

ASMI: Appendicular Skeletal Muscular Index; FFMI: Fat-Free Mass Index; FMI: Fat Mass Index; BCMI: Body Cell Mass Index; %TBW: Percentage Total Body Water; RFAI: Rectus Femoris Area Index; X/Y: Index Transversal axis (X)/anteroposterior axis (Y).

**Table 3 nutrients-14-04802-t003:** Difference in calorie and protein requirements and consumption between men and women.

	Total	Men	Women	*p*-Value
Calories Requirement (kcal/day)	1772 (178.12)	1894 (149)	1650 (107)	<0.01
Calories Consumption (kcal/day)	1364 (417)	1333 (455)	1433 (410)	0.40
Calories Consumption (%)	78.76 (16.88)	70.33 (22.83)	86.87 (23.98)	0.01
Protein Requirements (g/day)	79.26 (16.88)	87.82 (12.33)	70 (16.61)	<0.01
Protein Consumption (g/day)	1.15 (0.44)	1.07 (0.41)	1.22 (0.47)	0.23
Protein Consumption (%)	88.58 (34.20)	81.81 (31.27)	94.13 (36.53)	0.23

**Table 4 nutrients-14-04802-t004:** Difference in metabolic parameters depending on the presence of diabetes mellitus (DIABETES) or alterations in carbohydrates metabolism (NO DIABETES).

	Diabetes	No Diabetes	*p*-Value
HbA1c (%)	6.86 (1.19)	6.03 (0.58)	<0.01
Glucose (mg/dL)	124.92 (38.19)	94.62 (19.38)	<0.01
Total cholesterol (mg/dL)	153.75 (37.13)	167 (39.95)	0.19
HDL cholesterol (mg/dL)	57.81 (32.50)	63.86 (23.60)	0.45
LDL cholesterol (mg/dL)	79.54 (29,66)	84.40 (23.93)	0.53
Tryglicerides (mg/dL)	108.81 (54.06)	81.79 (31.18)	0.03
Albumin (g/dL)	4.06 (0.56)	4.13 (0.38)	0.59
CRP/prealbumin	0.43 (0.51)	0.60 (0.89)	0.37

HbA1c: Glycated Hemoglobin.

**Table 5 nutrients-14-04802-t005:** Changes in macronutrients and micronutrients and their distribution before and 3 months after intervention.

	Start	3 Months	*p*-Value
Carbohydrates (g)	144.99 (45.68)	182.21 (54.83)	<0.01
Fiber(g)	12.53 (4.76)	17.39 (7.14)	<0.01
Proteins (g)	65.51 (21.01)	70.42 (25.34)	<0.01
Lipids (g)	60.54 (21.41)	68.70 (25.88)	0.03
SFA (g)	17.39 (8.42)	19.77 (9.51)	0.12
SFA (%TCV)	10.37 (8.2–14)	9.69 (7.16–12.48)	0.26
MUFA (g)	23.86 (10.99)	28.94 (12.51)	0.01
MUFA (%TCV)	15.35 (12.38–18.45)	14.28 (12.53–18.45)	0.55
PUFA(g)	6.17 (4.22)	8.44 (3.68)	<0.01
PUFA (%TCV)	3.39 (2.86–4.56)	4.09 (3.61–5.42)	<0.01
EPA (g)	0.08 (0.14)	0.22 (0,56)	0.12
DHA (g)	0.13 (0.20)	0.16 (0.22)	0.52
Cholesterol (mg)	300.85 (157.75)	301.71 (196.96)	0.37
Minerals			
Phosphorus (mg)	881.10 (338.67)	1116.27 (423.28)	<0.01
Magnesium (mg)	151.65 (60.31)	202.09 (79.43)	<0.01
Calcium (mg)	708.59 (327.23)	982.12 (377.23)	<0.01
Iron (mg)	7.71 (3.09)	10.28 (4.17)	<0.01
Zinc (mg)	6.64 (3.25)	8.09 (3.66)	<0.01
Sodium (mg)	1569.35 (845.39)	1742.61 (824.38)	0.12
Potassium (mg)	1943.22 (687.02)	2205.49 (824.09)	0.04
Iodine (mg)	30.83 (25.16)	60.63 (33.64)	<0.01
Selenium (mg)	35.39 (24.01)	50.66 (27.62)	0.01
Copper (mg)	0.77 (0.54)	0.93 (0.49)	0.08
Vitamins			
Vitamin A (IU)	1152.26 (1263.39)	1347.76 (973.54)	0.40
Vitamin B1 (mg)	0.89 (0.46)	1.04 (0.72)	0.20
Vitamin B2 (mg)	1.19 (0.59)	1.51 (0.69)	<0.01
Niacin (mg)	11.96 (6.85)	13.84 (6.77)	0.10
Vitamin B5 (mg)	0.14 (0.42)	1.22 (1.00)	<0.01
Vitamin B6 (mg)	1.17 (0.61)	1.43 (0.68)	0.01
Folic Acid (mg)	138.75 (75.32)	174.29 (88.33)	0.01
Vitamin B12 (mg)	5.54 (7.88)	5.26 (4.50)	0.83
Vitamin C (mg)	96.55 (67.59)	109.29 (71.45)	0.33
Vitamin D (mg)	4.02 (6.41)	4.96 (5.57)	0.40
Vitamin E (mg)	5.77 (3.33)	7.62 (3.89)	<0.01
Vitamin K (mg)	1.52 (5.97)	16.93 (15.81)	<0.01

SFA: Saturated Fatty Acids; MUFA: Monounsaturated Fatty acids; PUFA: Polyunsaturated Fatty Acids; EPA: eicosapentaenoic acid; DHA: docasahexaenoic acid; %TCV: percentage total caloric value.

**Table 6 nutrients-14-04802-t006:** Changes in anthropometry, body composition and muscle function according to gender at baseline and 3 months after nutritional intervention.

	Men			Women		
**Anthropometry**						
	Baseline	3 Months	*p*-Value	Baseline	3 Months	*p*-Value
%Weight Loss	10.05 (7.03)	−0.25(5.57)	<0.01	12.84 (13.04)	−0.72 (4.95)	<0.01
BMI (kg/m^2^)	25.53 (4.30)	24.72 (4.04)	0.21	23.77 (6.18)	23.11 (5.56)	0.53
Arm circumference (cm)	25.43 (2.53)	25.67 (2.77)	0.32	23.99 (4.21)	24.16 (3.91)	0.65
Calf Circumference (cm)	32.75 (3.11)	33.33 (2.75)	0.16	24.16 (3.91)	30.63 (3.81)	0.18
Handgrip Strength						
Handgrip Strength (kg)	23.81 (7.61)	24.03 (8.81)	0.44	14.77 (6.66)	15.13 (5.69)	0.95
Bioelectrical Impedanciometry						
Resistance (ohm)	501 (76)	502 (85)	0.95	588 (86)	586 (87)	0.83
Reactance (ohm)	44.61 (9.43)	45.86 (11.43)	0.51	49.10(9.01)	49.43(11.29)	0.82
Phase Angle (°)	5.11 (0.89)	5.24 (1.15)	0.42	4.78 (0.66)	4.81 (0.79)	0.84
ASMI (kg/m^2^)	7.11 (0.91)	7.15 (0.94)	0.71	5.79 (0.91)	5.81 (0.94)	0.71
FFMI (kg/m^2^)	18.35 (2.76)	18.34 (2.80)	0.98	16.65 (3.20)	16.29 (2.19)	0.46
FMI (kg/m^2^)	6.80 (2.15)	6.72 (2.25)	0.76	6.98 (4.10)	6.85 (4.07)	0.51
BCMI (kg/m^2^)	8.97 (1.74)	9.09 (2.00)	0.48	7.64 (1.37)	7.67 (1.40)	0.79
%TBW	58.18 (4.13)	58.28 (4.83)	0.88	53.75 (11.35)	55.91 (7.17)	0.32
Rectus Femoris Ultrasonography						
RFAI (cm^2^/m^2^)	1.36 (0.55)	1.31 (0.57)	0.31	1.18 (0.35)	1.14 (0.37)	0.19
X/Y (cm^2^/m^2^)	3.39 (1.56)	3.55 (1.48)	0.46	3.79 (1.57)	3.56 (1.24)	0.46

ASMI: Appendicular Skeletal Muscular Index; FFMI: Fat-Free Mass Index; FMI: Fat Mass Index; BCMI: Body Cell Mass Index; %TBW: Percentage Total Body Water; RFAI: Rectus Femoris Area Index; X/Y: Index Transversal axis (X)/anteroposterior axis (Y).

**Table 7 nutrients-14-04802-t007:** Difference in the metabolic parameters before and 3 months after the start of the nutritional intervention based on the diagnosis of diabetes (DIABETES) or alterations in the carbohydrate’s metabolism (NO DIABETES).

	Diabetes			No Diabetes		
	Baseline	3 Months	*p*-value	Baseline	3 Months	*p*-Value
HbA1c (%)	6.87 (1.24)	7.18 (1.09)	0.02	6.05 (0.60)	6.10 (0.62)	0.30
Glucose (mg/dL)	123.56 (38.79)	131.38 (29.76)	0.10	94.62 (19.39)	89.83 (20.03)	0.55
Total cholesterol (mg/dL)	154 (37.01)	158 (38.99)	0.29	167 (39.95)	172 (46.22)	0.49
HDL cholesterol (mg/dL)	58 (33.48)	60.94 (27.87)	0.25	65.50 (23.94)	63.7 (20.79)	0.34
LDL cholesterol (mg/dL)	79.45 (30.55)	83.62 (31.35)	0.29	88.83 (19.56)	95.83 (43.66)	0.45
Tryglicerides (mg/dL)	109.89 (55.15)	105.34 (50.23)	0.26	81.79 (31.18)	86.92 (29.17)	0.32
Albumin (g/dL)	4.09 (0.53)	4.25 (0.42)	0.02	4.13 (0.38)	4.02 (0.42)	0.14
CRP/prealbumin	0.38 (0.51)	0.70 (2.07)	0.38	0.47 (0.74)	0.26 (0.38)	0.16

## Data Availability

Not applicable.

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
