# Peer review of "Real World Practice Study of the Effect of a Specific Oral Nutritional Supplement for Diabetes Mellitus on the Morphofunctional Assessment and Protein Energy Requirements"

_nutrients, 2022, doi:10.3390/nu14224802_

Round 1
Reviewer 1 Report
The manuscript may be of interest, especially in clinical work and research. However, I have the following comments.
II. Major comments:
1. I suggest including a figure that shows the total number of patients who were selected in the study, and what happened to them (how many continued or dropped out).
2. Correct the wording of the manuscript. Detect grammatical errors. Also, some words are written in Spanish
Table 2, años and dinamometria.
3. Authors report energy intake as fat, protein, and carbohydrate. But the formula used did not provide vitamins or minerals?
4. The introduction is good, but it is necessary to include background information regarding the role of nutrients in metabolic control and body composition of patients with DM.
5. Why were no significant changes observed in TAG? Discuss this point.
6. Regarding total lipids, in what % were they distributed (SFA, MUFA, n-6 and n-3 PUFAs)
7. The discussion is very general, it is necessary to include nutritional and metabolic aspects that allow a better understanding of the observed effects. Especially glycemic control. I even suggest including mechanistic aspects (example: use of energy substrates and muscle mass).
8. What future projections and applications do you have for this type of intervention?
8. Finally, the results are interesting, but the manuscript requires a major revision, especially in the structure, writing and discussion of metabolic and nutritional aspects.
I. Minor Comments:
1. Improve the wording of the objective of the study
2. Replace "nutritional survey" with "nutritional questionnaire".
3. I suggest editing the wording of the manuscript. The authors use too many sentences in a Spanish-English
4. Replace "p-valUE" with "p-value"
5. Don't use "when"
Author Response
Dear reviewers and editorial office:
First, I would like to thank you for the trust placed in our group by reviewing and considering our article.
According to the comments received, we have made a series of corrections in our article that I list below:
REVIEWER 1:
The manuscript may be of interest, especially in clinical work and research. However, I have the following comments.
- Major comments:
- I suggest including a figure that shows the total number of patients who were selected in the study, and what happened to them (how many continued or dropped out).
The reviewer was right about the lack of a flow chart of allocation. We have added this flowchart with CONSORT model.
- Correct the wording of the manuscript. Detect grammatical errors. Also, some words are written in Spanish
Table 2, años and dinamometria.
Sorry for the mistake. We have corrected this words and we have revised english throughout the manuscript.
- Authors report energy intake as fat, protein, and carbohydrate. But the formula used did not provide vitamins or minerals?
We have added vitamins and minerals. We also added the type of fatty acids and carbohydrates content in formula.
- The introduction is good, but it is necessary to include background information regarding the role of nutrients in metabolic control and body composition of patients with DM.
We have included some sentences about ONS and glycemic control, metabolic parameters and body composition. “In patients with diabetes and malnutrition, nutritional intervention for caloric in-crease can be associated with a worsening of glycemic control. This is the reason to use a diabetes-specific oral nutritional supplements. These formulas can have the following characteristics: a) the reduction of energy from carbohydrates and replace it with energy from lipids; b) use of carbohydrates with a low glycemic index such as lactose or iso-maltulose; or c) increase the amount of soluble fiber to decrease glucose absorption [14]. These types of formulas usually have a high percentage of protein to improve nutri-tional status and muscle function. On the other hand, these formulas are usually en-riched in monounsatured (MUFA) and polyunsatured fatty acids (PUFA) to find a benefit in the lipid profile of these patients [15].”
- Why were no significant changes observed in TAG? Discuss this point.
We have added this sentence to discussion: “Despite slight increase in glycated hemoglobin, it wasn’t seen any changes on tryglic-erides. This fact could be related to the increase on lipids with MUFA and PUFA consumption despite the increase of carbohydrate consumption.”
- Regarding total lipids, in what % were they distributed (SFA, MUFA, n-6 and n-3 PUFAs)
We have added a table with changes in lipids and its distribution (table 4). We also have added the distribution of fatty acids in intervention formula.
- The discussion is very general, it is necessary to include nutritional and metabolic aspects that allow a better understanding of the observed effects. Especially glycemic control. I even suggest including mechanistic aspects (example: use of energy substrates and muscle mass).
We have added some paragraphs about:
- Glycemic control and metabolic aspects: “A slight increase in glycated haemoglobin and plasma albumin was observed in patients with diagnosed diabetes mellitus. This may be related to the increase in the consumption of carbohydrates after the decrease in the initial intake in relation to the underlying pathology. This data differs from other data observed in studies with specific formulas in diabetes in which an improvement in glycaemic parameters was ob-served, although in these studies a medical nutrition therapy with complete enteral nutrition was usually evaluated [24]. In other studies, in which oral nutritional supplements were used, they were used as a substitute for meals and not in patients with malnutrition, so the results are not comparable with our population [27]. However, in the sample studied, no alterations were observed in basal glycaemia or in lipid param-eters, neither in patients with diabetes nor in patients with prediabetes. Despite slight increase in glycated hemoglobin, it wasn’t seen any changes on tryglicerides. This fact could be related to the increase on lipids with MUFA and PUFA consumption despite the increase of carbohydrate consumption [28]. Albumin is an imprecise biomarker that can be interfered by many situations as inflammation and hydration state of the patient. The change of this parameter in our sample is unspecific and there is no easy explanation. More specific nutritional biomarkers as PCR/prealbumin ratio did not show differences but their use is promising to evaluate the prognosis, specially in patients with acute pathologies [10].”
- Body composition: “The adequate adjustment of the diet in the patient with malnutrition is basic. If the requirements cannot be achieved with an adaptation of the diet, the use of artificial supplementation is necessary to meet these requirements. In our sample, an adequate range of caloric-protein requirements was observed with the nutritional intervention. However, in patients with diabetes it is also necessary to achieve an adequate glycemic control, because it has been observed that poorer glycemic control is associated with a greater decline of muscle mass [7]”.
- The use of specific oral nutritional supplementation for diabetes has shown in different studies a better postprandial glycemic control evaluated after its intake [14,30]. Other studies with meal-replacement plan during a short period of time have shown an improvement in the nocturnal glycemic profile [28]. Nevertheless, these interventions are not in a complete diet planning, and it has not been used in patients with disease related malnutrition. In this context, it can influence the type of prescribed diet, the underlying disease and adherence to oral nutritional supplementation.
- What future projections and applications do you have for this type of intervention?
We have discussed the possible lines of investigation in base of this study: “This study allows us to propose new lines of research on the use of diabetes-specific nutritional supplementation, with a control group and in specific groups of patients at nutritional risk. The use of the different morphofunctional assessment techniques must be basic in all nutritional assessment studies, given that we increas-ingly have more techniques that can be used in our daily clinical practice, such as bi-oimpedance measurement and nutritional ultrasound.”
We have also commented the strengths of study with possible uses of the results of this: “The main strengths of this study are the evaluation of diabetic-specific enteral in the form of oral nutrition supplement associated with diet in real clinical practice, since there are not many studies that evaluate this method of medical nutrition therapy. This fact allows us to extrapolate our results to generalized clinical practice. On the other hand, the evaluation from a morphofunctional point of view in this type of patients al-lows us a complete assessment and allows us to monitor the different spheres of nutri-tional status (evaluation of intake, anthropometry, body composition and muscle func-tion) and to be able to personalize most appropriate way of treatment”.
- Finally, the results are interesting, but the manuscript requires a major revision, especially in the structure, writing and discussion of metabolic and nutritional aspects.
We have made the changes in structure, language, and content in the different sections as you proposed.
- Minor Comments:
- Improve the wording of the objective of the study
We have changed the last paragraph of introduction and rewritten the objective.
Line - : For this reason, a real-life study is proposed to describe the effect of a diabe-tes-specific oral nutrition supplementation in patients with disease related malnutri-tion. The main objective of this study was to prove the influence of Medical Nutrition Therapy with a specific oral nutritional supplementation through morphofunctional assessment in patients with malnutrition and diabetes or altered metabolism of car-bohydrates.
- Replace "nutritional survey" with "nutritional questionnaire".
As you suggest, we have changed survey for questionnaire.
- I suggest editing the wording of the manuscript. The authors use too many sentences in a Spanish-English
We have revised all the document and changed the words and expressions that could be wrong.
- Replace "p-valUE" with "p-value"
We have changed this mistake.
- Don't use "when"
We have changed the expression with this adverb.

Reviewer 2 Report
The authors address in a preliminary study the problem of malnutrition in a clinical population of patients with DM/Pre-DM and treatment with an oral supplement.
Comments:
Extensive English editing is needed. Examples of sentences needing correction are:
Line 13, 79 (arm is preferred to branch)
14 (supplementation of diabetes is not correct)
31, 31 entity is not the correct word
36
84
73-74 Morphofunctional Assessment... needs a citation
99, 122 "artificial" can be removed
138
Also check for proper plurality and use of articles.
Methods:
Line 118. Describe the nutrition education provided.
120: what are food enrichment "measures?"
Describe the type of carbohydrate in the supplement (table 1).
How was compliance with the prescribed amount of supplement monitored? Provide data on the % of patients consuming, for example, 100%, 50%... of the prescribed amount.
You mention % weight loss as an outcome but do not describe collecting these data prior to the intervention. How did you know the rate of weight loss prior to the intervention?
Results.
Provide a more complete nutritional analysis of the diet at baseline and month 3 in a table. Also, it would be valuable to know the % of kcals and protein consumed from the supplement vs from the diet. Did the patients reduce their kcal intake from food to compensate for the supplement? Some weight gain was seen, so perhaps not.
Table 2, 3: Check "P-value" column heading.
Did any participants drop-out before month 3?
Discussion:
293: "Hydrocarbon" should be replaced with carbohydrate, lipid.
302-303: provide citations.
Serum albumin is an imprecise, outdated biomarker of nutritional status. Address this in the discussion.
392: "light" is not correct.
Author Response
Dear reviewers and editorial office:
First, I would like to thank you for the trust placed in our group by reviewing and considering our article.
According to the comments received, we have made a series of corrections in our article that I list below:
REVIEWER 2:
The authors address in a preliminary study the problem of malnutrition in a clinical population of patients with DM/Pre-DM and treatment with an oral supplement.
Comments:
Extensive English editing is needed. Examples of sentences needing correction are:
We have english vocabulary and grammar throughout the manuscript.
Line 13, 79 (arm is preferred to branch)
We have changed this word as the reviewer proposes.
14 (supplementation of diabetes is not correct)
We have changed this mistake in all the text with “diabetes-specific oral nutritional supplementation”.
31, 31 entity is not the correct word:
We have changed this word in text. “Disease-related malnutrition (DRM) is an pathology with a high prevalence, reaching up to 60% in hospitalized patients with chronic diseases [1]. This malnutrition is more striking in elderly patients and is closely related to sarcopenia, another highly prevalent disease in elderly patients.”
36
We have changed this sentence: “In fact, in institutionalized diabetic patients over 65 years, it has been observed that 21.2% are malnourished and that 39.1% are at risk of malnutrition [2].”
84
We have changed this sentence and wording of objective: “For this reason, a real-life study is proposed to describe the effect of a diabe-tes-specific oral nutrition supplementation in patients with disease related malnutri-tion. The main objective of this study was to prove the influence of Medical Nutrition Therapy with a specific oral nutritional supplementation through morphofunctional assessment in patients with malnutrition and diabetes or altered metabolism of car-bohydrates.”
73-74 Morphofunctional Assessment... needs a citation.
We have added a citacion to article. García Almeida JM, García García C, Vegas Aguilar IM, Bellido Castañeda V, Bellido Guerrero D. Morphofunctional assessment of patient´s nutritional status: a global approach. Nutr Hosp 2021; 38:592–600. https://doi.org/10.20960/nh.03378
99
We have changed this paragraph: “An exhaustive anamnesis was carried out on affiliation data, personal history, evolution of the disease and nutritional history. It was performed a classic anthropo-metric evaluation, bioelectrical impedanciometry and muscle ultrasound evaluation. Analysis with nutritional parameters was requested according to usual clinical practice.”
122 "artificial" can be removed
We have removed the term and change the expression for Oral Nutritional Supplementation
138
We have adapted the paragraph: “The energy expenditure of the patients was determined by means of the Har-ris-Benedict Equation multiplied by a Stress Factor of 1.3 and the protein require-ments were determined by means of the factor 1-1.5 g of protein per kilogram of the patient's adjusted weight. We based requirements on the patient's clinical situation and comorbidities as the recommendations made by the clinical guidelines of the European Society for Clinical Nutrition and Metabolism in surgery and oncology suggests. This decision was made because most of the patients had underlying oncological and/or surgical pathology [16,17].”
Also check for proper plurality and use of articles.
We have checked all document to correct this.
Methods:
Line 118. Describe the nutrition education provided.
We described nutritional education in this sentence: “Patients received nutritional education with a dietitian in adaptation of oral diet to increase protein-energy intake and they received education in consumption of oral nutritional supplementation. The adherence of these diets was assessed each fourteen days with a phone call by a dietitian to improve compliment of the calorie restriction and macronutrient distribution. The diet compliance was verified with a telephone dietary questionnaire every fourteen days and a four-days dietary ques-tionnaire of in face-to-face visits”.
120: what are food enrichment "measures?"
We referee to a diet with an increase in calories and proteins. We have adapted text to understand the sentence. “ Patients received education on adapted oral diet to increase calories and protein in patients with diabetes or carbohydrate metabolism disorders(prediabetes)”.
Describe the type of carbohydrate in the supplement (table 1).
We have added the type of carbohydrate in table 1. We have added type of fatty acids in formula.
How was compliance with the prescribed amount of supplement monitored? Provide data on the % of patients consuming, for example, 100%, 50%... of the prescribed amount.
We have added the compliance to supplement in text: “56 (93.3%) patients consumed 100% of oral nutritional supplementation prescribed; 1 (1.7%) consumed 50%; 1 (1.7%) consumed 25% and 2 (3.3%) patients consumed no supplementation.”
You mention % weight loss as an outcome but do not describe collecting these data prior to the intervention. How did you know the rate of weight loss prior to the intervention?
We have added the anthropometric measures to methods: “Anthropometry: The anthropometric variables measured were weight (kg); height (meters); body mass index (BMI) (weight/height ∗ height) (kg/m2); arm circumfer-ence (AC); and calf circumference (CC). The percentage of weight loss was calcu-lated: Start Weight Loss = ((Usual weight (kg)- Present weight (kg))/Usual weight)*100; 3 Months Weight Loss = ((Initial weight (kg) – 3 months weight)/Initial weight)*100.”
Results.
Provide a more complete nutritional analysis of the diet at baseline and month 3 in a table. Also, it would be valuable to know the % of kcals and protein consumed from the supplement vs from the diet. Did the patients reduce their kcal intake from food to compensate for the supplement? Some weight gain was seen, so perhaps not.
We have added table 4 with all changes in nutritional questionnaire before and 3 months after intervention.
The nutrition questionnaire was made including the supplement, but we didn’t separate what nutrients are from supplement and what nutrients are from natural oral diet.
Table 2, 3: Check "P-value" column heading.
We have checked this word. Sorry for the inconvenience.
Did any participants drop-out before month 3?
We have added a flow chart to show the loss of follow-up.
Discussion:
293: "Hydrocarbon" should be replaced with carbohydrate, lipid.
This term is wrong, so we have changed throughout the manuscript.
302-303: provide citations.
In this sentence we summarize the main results of our study so we can’t provide citations from that. However, we have completed the sentence to clarify this point.
Serum albumin is an imprecise, outdated biomarker of nutritional status. Address this in the discussion.
We have added this idea to discussion: “Albumin is an imprecise biomarker that can be interfered by many situations as in-flammation and hydration state of the patient. The change of this parameter in our sample is inespecific and there is no easy explanation. More specific nutritional bi-omarkers as PCR/prealbumin ratio did not show differences but their use is promising to evaluate the prognosis, specially in patients with acute pathologies [10].”
392: "light" is not correct.
We have corrected the term.

Round 2
Reviewer 1 Report
The authors report the changes made, but cannot identify all those changes in the manuscript. However, I maintain the comments made in my first evaluation:
I. Major Comments:
1. Send a version of the manuscript with the changes highlighted (example: red color)
2. The figures help to improve the understanding of the manuscript, but they must improve the quality of the figures (small print or numbers that cannot be easily identified or visualized).
Author Response
The authors report the changes made, but cannot identify all those changes in the manuscript. However, I maintain the comments made in my first evaluation:
I. Major Comments:
- Send a version of the manuscript with the changes highlighted (example: red color)
We have sent a version with all corrections suggested by reviewer in another color. It could be some changes from the initial version, due to other reviewer and Academic editor suggestion. Sorry for the incovenience.
2. The figures help to improve the understanding of the manuscript, but they must improve the quality of the figures (small print or numbers that cannot be easily identified or visualized).
We have tried to improve quality. We have sent an archive with figures and original .ppt with graphs.
Reviewer 2 Report
Table 1. "Chrome," "Molybdemun," "Coline," "Vitamina" and "Cupper" are not correct.
Use English terms in table 4, figure 1 and throughout the paper. The authors stated that they did so in this revised version, but they did not.
Line 278: These bottle amounts were prescribed; the next lines describe consumption rates. It seems inappropriate to include in final analysis those participants with <50% compliance rates. Why did you include them?
Line 301-: Comment on the intakes of micronutrients at baseline and follow-up. Line 298 is not proper English.
Table 5 shows p-valor.
Figure 4 needs to show p values.
Line 417: remove "artificial"
Lines 433, 436, 440: improper English
Author Response
Dear reviewers and editorial office:
First, I would like to thank you for the trust placed in our group by reviewing and considering our article.
According to the comments received, we have made a series of corrections in our article that I list below:
REVIEWER 2 (Round 2):
Table 1. "Chrome," "Molybdemun," "Coline," "Vitamina" and "Cupper" are not correct.
Sorry for the typographic errors. I didn’t notice them.
Use English terms in table 4, figure 1 and throughout the paper. The authors stated that they did so in this revised version, but they did not.
Sorry for the inconvenience. I have changed all this errors.
Line 278: These bottle amounts were prescribed; the next lines describe consumption rates. It seems inappropriate to include in final analysis those participants with <50% compliance rates. Why did you include them?
The analysis of the adherence is at 3 months, but these patients consumed total supplementation more than a half of time of intervention. We did not consider adequate exclude these patients in order to lose representativity from the sample.
We have include in statistical analysis this plan: "It was made an intention-to-treat analysis of patients who consumed supplementation more than a half time of intervention."
Line 301-: Comment on the intakes of micronutrients at baseline and follow-up.
We have added the suggestion.
“An improvement in consumption of Monounsaturated Fatty acids and Polyun-saturated Fatty acids after the intervention was observed (table 4). In the same way, an improvement in consumption of minerals except sodium and copper; and in consumption of vitamins except vitamin A, B1, B3, B12, C and D was observed (table 4)”.
Line 298 is not proper English.
“Table 4: Changes in macronutrients and micronutrients and their distribution be-fore and 3 months after intervention”.
Table 5 shows p-valor.
We have changed it.
Figure 4 needs to show p values.
We have added p-value.
Line 417: remove "artificial"
We have removed this term throughout the text.
Lines 433, 436, 440: improper English
We have changed the sentences:
- 433: “The use of diabetes-specific oral nutritional supplementation has shown in different studies a better postprandial glycemic control”
- 436: “Other studies with meal-replacement plan during a short period of time have shown an improvement in the glycemic profile [28]. These interventions are not in a complete diet planning, and it has not been used in patients with disease related malnutrition.”
- 440: “The main strengths of this study were the evaluation of diabetic-specific formula as oral nutritional supplement associated with diet in real clinical practice”
